



# Comments on "Ideas and perspectives: is shale gas a major driver of recent increase in global atmospheric methane?" by Robert W. Howarth (2019)

Michael D. Lewan[1]

5   [1]U.S. Geological Survey, Emeritus, Box 25046, MS 977, Denver Federal Center, Denver, Colorado 80225, USA

*Correspondence to:* Michael D. Lewan (mlewan@usgs.gov)

**Abstract.** The ideas and perspectives presented by Howarth (2019) on shale gas being a major cause of recent

increases in global atmospheric methane are based on his notion that stable carbon isotopes of methane ($\delta^{13}C_1$) of

10   shale gas are lighter than that of conventional gas based on a meager and unrepresentative data set.   A plethora of

publicly available data show that the $\delta^{13}C_1$ values of shale gas are typically heavier than those of conventional gas.

This contradiction renders his ideas, perspectives, and calculations on methane emissions from shale gas invalid.

## 1 Main Text

15   Methane as a greenhouse gas in the atmosphere can be a contributor to climate change. As well referenced in the

subject paper (Howarth, 2019; Figure 1), global atmospheric methane concentrations rapidly increased from 1980 to

1991 at a rate of ~13 ppb/yr, increased more slowly from 1991 to 1999 at a rate of ~5 ppb/year, and then remained

essentially constant from 1999 to 2006. Concentrations began increasing again from 2006 to 2015 at a rate of ~6

ppb/yr, which is similar to that observed from 1991 to 1999 and lower than that from 1980 to 1991 by a factor of two.

The stable carbon isotope ratio ($\delta^{13}C_1$) of atmospheric methane became lighter ($^{13}C$ depleted) starting in 2008 with the

most recent increase in atmospheric methane concentrations. As correctly noted in the subject paper, previous studies

by Schaefer et al. (2016) and Schwietzke et al. (2016) concluded that the recent increase in methane concentrations

are a result of increasing microbial methane emissions with lesser to no significant increased contributions from fossil-

fuel emissions.


The subject paper questions these findings by maintaining that the $\delta^{13}C_1$ of shale gas is notably lighter than that of

conventionally produced fossil-fuel gas.  This is shown in Figure 3b of the subject paper with $\delta^{13}C_1$ means of -40.0

‰ for "fossil fuels other than shale gas" and -46.9‰ for shale gas.  The subject paper dismisses the large and well-

documented data set of Sherwood et al. (2017) on the grounds that "some" of the $\delta^{13}C_1$ values for shale gas of the 647

shale-gas entries representing 17 basins, 56 fields, and 41 formations are not representative of shale gas.  It is unclear

why Howarth (2019) did not use an edited version of the extensive data base by Sherwood et al. (2017) and instead

uses the mean of three meager data sets (i.e., Bakken, Utica, and Barnett) that are not representative of shale gas.

The Bakken is not a shale-gas play as clearly stated in the first sentence of Schoell et al. (2011), which the subject

paper designates as shale-gas.  The Bakken Shale is a shale-oil play in which oil and minor associated gas are generated





from the upper and lower Bakken Shale that migrates into clastic and carbonate lithologies of the middle of the Bakken Shale, which has been the focus of horizontal drilling for oil (Gaswirth and Marra, 2015). Surprisingly, this migration element of the Bakken shale play is the one specific concern Howarth (2019) has with some of the entries in the data base by Sherwood et al. (2017). Howarth (2019) gives a mean Bakken $\delta^{13}C_1$ value of -47.0‰ and cites Schoell et al.

(2011) as one of the three data sets averaged to obtain the mean shale-gas value (i.e., -46.9‰). However, Schoell et al. (2011) report $\delta^{13}C_1$ values for only 8 gases that are associated with produced oil from the Bakken Formation at various levels of thermal maturity. All eight of these gases have $\delta^{13}C_1$ values heavier than -47.0‰, which makes the cited mean unattainable. A calculated mean of these eight values is -46.0 ± 0.74‰ (Table 1).

Botner et al. (2018) do an excellent job of reporting on the $\delta^{13}C_1$ values of methane dissolved in ground waters in Ohio where hydraulic-fracturing of the Utica Shale is occurring. They prescribe two $\delta^{13}C_1$ values for natural gas (conventional and shale gas in their Figure 3) for a visual comparison showing that the methane dissolved in the groundwaters is independent of hydraulic-fracturing activity. Presumably, their conventional gas value of -41.3‰ was collected from an abandoned gas well through an intersecting water well, but the source of their shale gas value

of -47.3‰ is not given. It is this single $\delta^{13}C_1$ value that Howarth (2019) uses to calculate his mean shale-gas value (i.e.,-46.9‰). This is unfortunate in light of the extensive $\delta^{13}C_1$ data reported by Burruss and Laughey (2010) on mostly unconventional gas sourced by the Utica Shale in the Appalachian region. These authors classified the collected gases with respect to whether they were associated or unassociated with oil production. Summary of these analyses in Table 1 differentiated between the two with one gas produced from the Utica "*proper*" with a $\delta^{13}C_1$ value

of -27.0‰, and 39 gases that have migrated out of the Utica into adjoining rock units with a mean $\delta^{13}C_1$ value of - 30.9‰. Also given, is a mean $\delta^{13}C_1$ value of -38.7‰ for 23 gases associated with oil production at lower thermal maturities. These heavier values for the Utica shale gas do not support the prescribed single Utica value used by Howarth (2019; -47.3‰).

The Barnett data set from Townsend-Small et al. (2015) used in the subject paper is also inappropriate because the mean $\delta^{13}C_1$ value of -46.5 ‰ includes only atmospheric samples and not well-head samples of produced shale gas that have a mean value of -41.0 ± 2.6‰ (132 samples, Zumberge et al., 2012). It is this isotopically heavier ($^{12}C$ depleted) mean value from well-heads collected over five counties that should be used as an example of Barnett shale gas and not atmospheric-gas samples from gas-well pads that can contain atmospheric microbial methane. The futility

of the notion by Howarth (2019) that $\delta^{13}C_1$ values of shale gas are lighter than conventional gas is also shown in the Barnett gas data reported by Rodriguez and Philp (2010). They characterize their gases into two groups. Group 1 gases are methane-rich (>95% $C_1$) and occur in the eastern more thermally mature part of the Fort Worth Basin (Vitrinote reflectans values greater than 1.2%$R_o$). Group 2 gases are wet (93 to 79% $C_1$) and occur in the western less thermally mature part of the basin (<1.2%$R_o$). As expected and shown in Table 1, the mean $\delta^{13}C_1$ for the more

thermally mature Group 1 gases (-38.5 ± 0.8‰) is heavier than that of the less thermally mature Group 2 gases (-44.5 ± 2.3‰ ). Like the Barnett gases reported by Zumberge et al. (2012), they are *proper* shale gases that have not



experienced migration out of their tight host rock. Figure 1 shows both groups have heavier $\delta^{13}C_1$ values than that prescribed by Howarth (2019) and that thermal maturity and not migration are responsible for their differences.

In addition to the Barnett being an unequivocal example of a shale-gas host, the Fayetteville of the Arkoma Basin and Marcellus of the Appalachian Basin are also unequivocal examples of major shale-gas hosts. Zumberge et al. (2012) report $\delta^{13}C_1$ values for shale gas produced from 98 wells in the Fayetteville over 5 counties in Arkansas with a mean of -38.2 ± 1.5‰. This mean is significantly heavier than the mean shale gas $\delta^{13}C_1$ (-46.9‰) prescribed by Howarth (2019). Similarly, the mean of 1,502 shale gases from mud-gas logging (MGL) in the Marcellus proper also have a

significantly heavier isotopic signature with a mean $\delta^{13}C_1$ of -32.4 ± 3.8‰ (Table 1). It should be noted that gases from MGL have good one-to-one correlations with produced gases (Weissenburger and Borbas, 2004; Dawson and Murray, 2011). Baldassare et al. (2014) present 682 $\delta^{13}C_1$ values collected during mud-gas logging that represent shale gas from overlying rock units sourced by the Marcellus. As shown in Table 1, mean $\delta^{13}C_1$ values get progressively lighter from the Marcellus (-32.4 ± 3.75‰) to the stratigraphically highest Catskill/Lochhaven

reservoirs (-42.1 ± 6.29‰). This is counter to the notion of Howarth (2019) that migration of shale gas results in isotopically heavier $\delta^{13}C_1$ values. In addition to natural data not supporting the migration fractionation of $\delta^{13}C_1$ proposed by Howarth (2019), it is not supported by experimental data as reported by Zhang and Kroos (2001). They state that diffusion of methane through water-saturated sedimentary rocks is most likely to cause fractionation of $\delta^{13}C_1$ during migration, but diffusion is not a major mode of gas migration in hydrocarbon systems. Their experiments at

subsurface conditions showed that the $\delta^{13}C_1$ of diffused methane is lighter than that of the methane source and not heavier as advocated by Howarth (2019).

   The heavier isotopic signature for shale gas is expected because the major shale-gas plays occur at higher thermal maturities with vitrinite reflectance values greater than 1.2%$R_o$. It is well established that with increasing thermal

maturity methane becomes the more dominant hydrocarbon and its $\delta^{13}C_1$ becomes heavier (Whiticar, 1994). Despite opposing data (Table 1), Howarth (2019) speculates that conventional methane becomes isotopically heavier ($^{13}C$ enriched) during migration as a result of $^{12}C$ being preferentially oxidized by "perhaps" bacteria using ferric iron or sulfate as the "oxidizing power". Unfortunately, the references he cites in this regard are not relevant. The papers by Whelan et al. (1986) and Rooze et al. (2016) are respectively concerned with anaerobic incubated shallow mud cores

and near surface sediments with labile organic matter, and not subsurface rocks with kerogen. The cited papers by Burruss and Laughrey (2010) and Hao and Zou (2013) discuss the possibility of ethane oxidation and do not consider oxidation of methane. Howarth (2019) does not note that Hao and Zou (2013) state that methane is the most stable petroleum compound and is not likely to be oxidized in the subsurface.

In addition to the $\delta^{13}C_1$ values prescribed for shale gas by Howarth (2019) not being representative, it is important to realize that $\delta^{13}C_1$ values are not a reliable parameter to differentiate shale gas from conventional gas on a global basis. As correctly stated and referenced by Howarth (2019), some shale gases have lighter $\delta^{13}C_1$ values (-50.7 to -53.3‰) but are not typical of major shale-gas plays. As shown in Figure 1, data from the cited references have lighter $\delta^{13}C_1$





values (Martini et al., 1998; McIntosh et al., 2002; Osborn and McIntosh, 2010), which is attributed to the addition of
microbial methane in shales that are at shallow depths or low thermal maturity (<1.2 %$R_o$) within sedimentary basins.
Theoretically, $\delta^{13}C_1$ values for shale gas can span the full range of values observed for conventional gases as reported
by Jenden et al. (1993) in Figure 1. However, major economic accumulations of shale gas occur in high thermal
maturity host rocks that have heavier $\delta^{13}C_1$ values. This adds considerable uncertainty in algebraic computations like
those of Howarth (2019) to estimate global contributions of methane emissions from shale gas. It should be noted
that direct measurements of Uunited States methane emissions between 2006 and 2015 by Lan et al. (2019) indicate
that despite an approximate 46% increase in gas production during this time period, total United States methane
emissions have remained essentially constant.

Table 1 gives the $\delta^{13}C_1$ means for compiled *"proper"* (-36.9 ± 6.3‰) and *"proper plus migrated"* (-36.5 ± 6.0‰)
shale gases. Similar to the approach of Howarth (2019), these means are not weighted by number of samples and are
referred to as unweighted. These two means are essentially the same but are significantly heavier than the $\delta^{13}C_1$ mean
of -46.9 ‰ shale-gas value prescribed by Howarth (2019). These heavier $\delta^{13}C_1$ values for shale gas are not unique to
the U.S. (Table 1 and Figure 1) with major shale gas from the Chinese Longmaxi Shale also having heavy $\delta^{13}C_1$ values
(-29.2 ± 1.2‰ mean of 76 samples; Feng et al., 2017). The heavier rather than lighter $\delta^{13}C_1$ values for shale gas have
a significant impact and cascading effect on calculated shale gas (SG), fossil-fuel except for shale gas ($FF_N$), and
biogenic gas ($B_N$) methane emissions according to the algebraic scheme used by Howarth (2019; Eqs. 1, 7, 8, and 12).
Howarth (2019) reduces his Eq. (7) for determining shale-gas emission (SG) using his prescribed $\delta^{13}C_1$ values for
fossil-fuel (-44.0‰), atmosphere (-53.5‰), biogenic (-62.5‰), and shale gas (-46.9‰). This results in his Eq. (8)
(i.e., Eq. (A)):

$$SG = -1.19FF_N + 19.4. \hspace{4cm} (A)$$

Using the same $\delta^{13}C_1$ values with the exception of the heavier and more representative mean $\delta^{13}C_1$ value of -36.9 for
shale gas (Table 1) in his Eq. (7), the expression of shale-gas emission becomes Eq. (B):

$$SG = -0.72FF_N + 11.86, \hspace{3.5cm} (B)$$

which has notably different coefficients than Eq. (A). Howarth (2019) proposes to derive $FF_N$ based on a series of
assumptions and shale-gas emission (SG) to determine methane emissions from fossil fuel excluding shale gas (i.e.,
$FF_N$) by his Eq. (12) (i.e., Eq. (C)) :

$$FF_N = 0.59SG + 2.9. \hspace{4cm} (C)$$

He substitutes his Eq. (12) (i.e., Eq. (C)) into his Eq. (8) (i.e., Eq. (A)) and solves for shale gas emissions (SG), which
results in a reported value of 9.4 Tg/yr. Substituting his Equation 12 into Eq. (B) based on the more representative



and heavier $\delta^{13}C_1$ value for shale gas (-36.9‰), the calculated shale-gas emission is 6.8 Tg/yr. Using this value in Eq. (12) (i.e., Eq. (C)) yields a methane emission from fossil fuel excluding shale gas of 6.9 Tg/yr instead of the 8.4 Tg/yr determined by Howarth (2019). These reductions resulting from using a more representative and heavier $\delta^{13}C_1$ value for shale gas also increases biogenic emissions from 10.6 to 14.7 Tg/yr in balancing the global increase of 28.4

Tg/yr reported by Worden et al. (2017) as given by Howarth (2019) in his Equation 1. Interestingly, an increase in biogenic methane emission with its isotopically lighter $^{13}C_1$ would better explain the decrease in atmospheric $\delta^{13}C_1$ since 2009 (Schaefer et al., 2016).

Sources and magnitude of methane emissions are important considerations in understanding the intricacies of climate
change. In this regard, it is critical that objective and cognizant science be presented on the issue. Howarth (2019) does not use representative shale-gas isotopic data, excludes a plethora of publicly available shale-gas data, does not acknowledge shale gas and conventional gas on a global basis cannot be readily distinguished based solely on $\delta^{13}C_1$ values, speculates contrary to field observations and laboratory experiments that migration causes conventional gases to have heavier $\delta^{13}C_1$ values than shale gas, does not consider the effects of thermal maturation on shale-gas $\delta^{13}C_1$
values, and neglects $\delta^{13}C_1$ data showing major shale-gas production is heavier and not lighter than conventionally produced gas. These numerous and significant shortcomings render his conclusions on global methane emissions from shale gas invalid.

**Competing interests**

The author declares that he has no conflict of interest.


**Acknowledgements and Disclaimers**

Comments and suggestions from Geoffrey Ellis (USGS) were especially helpful in preparing the original manuscript. Appreciation is extended to USGS editorial reviews by Janet Slate and Dave Ferderer. Thanks are extended to Carolyn Last (Edge Environmental) for bringing the subject paper to the author's attention. The author is grateful to Vincent
Nowaczewski's (Consulting geochemist) thorough review of the mathematic constructions in the subject paper and this Comment. Any use of trade, firm, or product names is for descriptive purposes only and does not imply endorsement by the US Government.

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





**Table 1** Compiled $\delta^{13}$C values for methane from various sources related to unconventional shale gas.
Italicized entries denote those presented in the subject paper by Howarth (2019).

| Gas Type | Mean (‰) | ± | 95% CI | Number | Reference |
|---|---|---|---|---|---|
| *Howarth (2019) Mean Values* | | | | | |
| *Fossil Fuels Excluding Shale ga.* | *-44.0* | *NG* | *0.55* | *NG* | *Howarth (2019)* |
| *Shale Gas* | *-46.9* | *0.40* | *0.45* | *3* | *Howarth (2019)* |
| *Utica (shale gas)*[a] | *-47.3* | *NA* | *NA* | *1* | *Howarth (2019)* |
| Utica Proper (shale gas) | -27.0 | NA | NA | 1 | Burruss and Laughery (2010) |
| Utica (migrated shale gas) | -30.9 | 3.16 | 0.99 | 39 | Burruss and Laughery (2010) |
| Utica (shale oil) | -38.7 | 2.33 | 0.95 | 23 | Burruss and Laughery (2010) |
| *Bakken (shale Oil)*[a] | *-47.0* | *NA* | *NA* | *8?* | *Howarth (2019)* |
| Bakken (shale Oil) | -46.0 | 0.74 | 0.51 | 8 | Schoell et al. (2011) |
| *Barnett (atmospheric gas)*[a] | *-46.5* | *1.70* | *1.70* | *34* | *Townsend-Small et al. (2015)* |
| Barnett Group 1 shale gas | *-38.5* | *0.80* | *0.04* | *15* | Rodriguez and Philp (2010) |
| Barnett Group 2 shale gas | *-44.5* | *2.30* | *0.80* | *35* | Rodriguez and Philp (2010) |
| Barnett shale gas | -41.0 | 2.60 | 0.44 | 132 | Zumberge et al. (2012) |
| Fayetteville shale gas | -38.2 | 1.50 | 0.30 | 98 | Zumberge et al. (2012) |
| Marcellus proper (MGL)[b] | -32.4 | 3.75 | 0.18 | 1,502 | Baldassare et al. (2014) |
| Catskill/Lochaven (MGL)[b] | -42.1 | 6.29 | 0.80 | 238 | Baldassare et al. (2014) |
| Brallier (MGL)[b] | -37.2 | 4.27 | 0.83 | 101 | Baldassare et al. (2014) |
| Geneseo (MGL)[b] | -34.6 | 3.33 | 1.06 | 38 | Baldassare et al. (2014) |
| Tully (MGL)[b] | -34.1 | 5.30 | 1.45 | 51 | Baldassare et al. (2014) |
| Hamilton (MGL)[b] | -33.3 | 3.44 | 0.42 | 254 | Baldassare et al. (2014) |
| Marcellus + migrated gas (MGL) | -35.6 | 4.40 | 0.79 | 2,184 | Baldassare et al. (2014) |
| Shale Gas (this study) | | | | | |
| Proper[c] | -36.9 | 6.29 | 5.04 | 6 | this study compilation |
| Proper + migrated[c] | -36.5 | 5.97 | 4.44 | 7 | this study compilation |

[a] used in unweighted sample mean for Shale Gas entry by Howartth (2019, Figure 3b)

[b] used in unweighted sample mean for Marcellus+ migrated gas

[c] unweighted sample means for this study entries.

MGL = mud gases collected during logging

NA = Not Applicable

NG = Not Given

CI = Confidence Interval



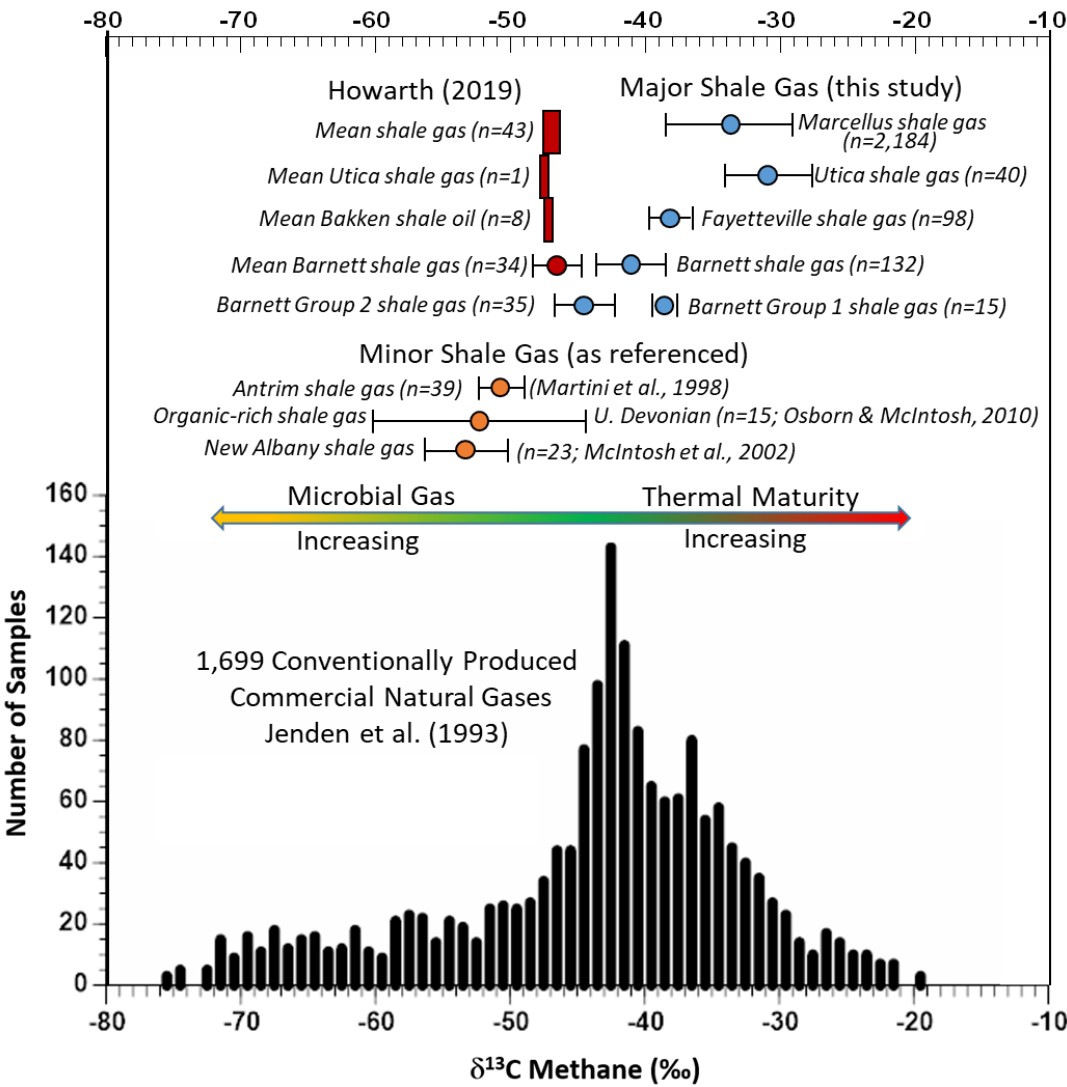

**Figure 1:** Upper half of figure compares mean and standard deviations of $\delta^{13}C_1$ values for major (blue symbols) and minor (orange symbols) shale-gas plays with the 95% confidence intervals (red rectangle symbols) and mean and standard deviations (red circled symbol) of shale gas as presented and referenced by Howarth (2019). Data and references are given in Table 1. Lower half of figure presents a $\delta^{13}C_1$ histogram for conventionally produced commercial natural gases (Jenden et al., 1993). Two-ended arrow in center of figure shows the general thermal-maturity and microbial-gas trends in $\delta^{13}C_1$ values based on Whiticar (1994).