# Peer review of "Comments on "Ideas and perspectives: is shale gas a major driver of recent increase in global atmospheric methane?" by Robert W. Howarth (2019)"

_Biogeosciences, 2019_

## Referee Comment (RC1) · Anonymous Referee #1 · 7 Mar 2020

The comment by Lewan rather aggressively dismisses the paper by Howarth (2019). However, the comment is not convincing. Before reaching this conclusion, I spent considerable time carefully reading through Lewan's comment, the original paper by Howarth (2019), the replies by Howarth to the reviews on the original discussion version of his paper, and also several of the papers and reports cited by Lewan. From this, I believe Lewan has mischaracterized the Howarth (2019) paper, and in many cases mischaracterizes the papers and reports he cites. Some of his criticisms seem exaggerated, and others are just plain wrong.

Cutting right to the chase, Lewan develops an estimate for the 13C in shale gas

methane that he believes is better than the estimate used by Howarth (2019), and then uses this to re-do the calculations of Howarth (2019). From this Lewan's comment concludes that the global increase in methane emissions from 2005-2015 was driven by increased emissions of 6.8 Tg per year from shale gas (72% of Howarth's estimate of 9.4 Tg per year), 6.9 Tg per year from other fossil fuels (82% of Howarth's estimate of 8.4 Tg per year), and 14.7 Tg per year from biological sources (39% greater than Howarth's estimate of 10.6 Tg per year). I find these relatively modest differences. In stark contrast, Lewan repeatedly speaks favorably of the paper by Schaefer et al. (2016), and concludes that his conclusions largely align with theirs. That is not in fact the case: Schaefer et al. concluded that the increase emissions from 2005-2015 were driven almost entirely from biological sources, stating that fossil fuel emissions may have actually declined. This is a qualitatively different finding.

The major criticism in Lewan's comment is that Howarth (2019) did not use representative values for the C13 signal of methane from shale gas. In the specific line-by-line comments below, I go through this argument in some detail. To summarize, Howarth specified that data should only be used for samples where it could be unambiguously demonstrated that the methane came from producing shale gas wells, and not from conventional gas wells or methane that had originated in shale but then migrated (where it might have been fractionated). The data presented by Lewan in his comment do not meet this standard, and in many cases the Howarth (2019) paper had already explained why. The replies by Howarth to reviewers on the submitted discussion-draft of his paper provided further explanation; see https://www.biogeosciences-discuss.net/bg-2019-131/bg-2019-131-AC1.pdf, https://www.biogeosciences-discuss.net/bg-2019-131/bg-2019-131-AC4.pdf, https://www.biogeosciences-discuss.net/bg-2019-131/bg-2019-131-AC2.pdf, and https://www.biogeosciences-discuss.net/bg-2019-131/bg-2019-131-AC3.pdf.

It is instructive that Lewan argues in his comment that natural gas produced from the Bakken fields is not "shale gas," and that he criticizes Howarth for stating otherwise.

Howarth (2019) clearly defines shale gas as that gas that is produced when methane and associated gases are released from being held tightly in shale rock by high-volume hydraulic fracturing. This is entirely consistent with the usage of the term by the Energy Information Agency (EIA) of the US DOE, upon whose data Howarth (2019) relied. And by this definition and the reporting by the EIA, the gas produced from the Bakken is shale gas. Lewan does not clearly state his definition of "shale gas," but based on the values he puts forward for 13C in methane, he includes as "shale gas" methane that has migrated from shale source formations to conventional gas reservoirs over geological time. This is not consistent with Howarth (2019) or the EIA.

Key to the Howarth (2019) presentation is the logic that the methane in conventional gas reservoirs is more enriched in 13C because of fractionation as this methane migrates over geological time to the reservoir. The fractionation, Howarth (2019) argues, is due to oxidation of some of the methane, with Fe(II) or sulfate serving as the electron acceptor. If so, then on average, to the extent shale was the source rock for the methane that has migrated, the methane in the conventional reservoir will be more 13C-enriched than that in the shale. Lewan's dismissal of this hypothesis is not convincing. For example, he cites one paper (Hao and Zou 2013) that stated methane is stable within the reducing environment of the shale; this is immaterial to the fate of methane migrating through sandstone where Fe(III) and sulfate may be present. The shale is a highly reducing environment compared to the sandstone.

The Lewan comment also criticizes Howarth (2019) for using a 13C value for methane in the air that reflects emissions from a producing shale gas well (from Townsend-Small et al. 2015). This criticism is misplaced, as it is precisely the 13C of the methane that reaches the atmosphere that is of interest. Of note, in a presentation at the December 2019 AGU meeting, Muehlenbachs and Arismendi reported that methane emissions to the atmosphere from gas development in western Canada were quite depleted in 13C (- 57 to -58) relative to the values used by Howarth (2019). They attributed this to a contribution from biogenic sources lying above the natural

gas resource: the gas development resulting in emissions that included these biogenic sources through the well casings that went through the biogenic-source areas (https://agu.confex.com/agu/fm19/meetingapp.cgi/Paper/520383). The point is that shale gas activity can result in methane emissions to the air that are even more depleted in 13C than is the methane in the shale gas actually be produced. This presentation occurred 6 months after Howarth (2019) was published and so of course not included in that analysis. Had these new results been included in Howarth (2019), it would have suggested an even greater increase in atmospheric methane emissions from shale gas development over the past decade.

Specific, detailed comments:

Abstract, lines 10-12: the comment does not in fact show that the data set used by Howarth (2019) is unrepresentative, and so these lines should be deleted.

Lines 21-24: the comment refers to Schaefer et al. (2016) and Schwietzke et al. (2016), but does not reference or refer to the subsequent paper by Worden et al. (2017). Worden et al. pointed out that the two earlier papers did not correctly consider the effect of changes in global biomass burning on atmospheric 13C in methane. When this is considered, Worden et al. (2017) concluded that the increase in methane emissions over the past decade was driven primarily by fossil fuel sources, not biological sources as Schaefer et al. and Schwietze et al. and stated. Lewan needs to add this further information to his comment, since it in fact supports the Howarth (2019) paper. Note that Howarth highlighted this Worden et al. work, so it is inexcusable for Lewan to ignore it.

Lines 26-31: the comment refers to the data set of Sherwood et al., and states "it is unclear why Howarth (2019) did not use an edited version of the extensive data base by Sherwood et al. (2017)." This is unfair, as Howarth directly addressed why he chose not to rely on this data set. Howarth (2019) stated: "some of the data listed as shale gas in that data set are actually for methane that has migrated from shale

to reservoirs (Tilley et al., 2011) and therefore may have been partially oxidized and fractionated (Hao and Zou, 2013). In other cases, the data appear to come both from conventional vertical wells and shale-gas horizontal wells in the same region, making interpretation ambiguous (Rodriguez and Philp, 2010; Zumberge et al., 2012). Note that in the Barnett shale region, Texas, the $\delta$13C ratio for methane emitted to the atmosphere ($-46.5$; Townsend-Small et al., 2015) is more depleted than the average for wells reported in the Sherwood et al. (2017) data set: $-44.8$ for "group 2A and 2B" wells and $-38.5$â$\breve{A}$Ľ‰ for "group 1" wells (Rodriguez and Philp, 2010) and a $-41.1$ average value (Zumberge et al., 2012)."

Howarth provided more detail on this in his reply to reviewer #3, who had specifically suggested the use of the Sherwood et al. data set, as well as data sources in a review paper by Tilley and Muehlenbachs (2013). In his reply (https://www.biogeosciences-discuss.net/bg-2019-131/bg-2019-131-AC3.pdf), Howarth wrote: "I followed the leads in the Tilley and Muehlenbachs(2013) review suggested by the reviewer, as well as those in the Sherwood et al. (2017) data set. With regard to the work cited by Tilley and Muehlenbach (2013), some of these studies refer to methane that has migrated from the original shale formation,and not to methane that would be released from shale through high-volume hydraulic fracturing (which is how I and most others define "shale gas"). Since my argument is that the methane would be subject to fractionation by partial oxidation during migration,it would not be appropriate to include data on these migrated gases. Included in the Tilley and Muehlenbach (2013) paper are data from Tilley et al. (2011): note that Hao and Zou (2013) specifically decided not to include those data in their modeling, noting that fractionation during migration seemed likely. Similarly, many of the samples listed by Sherwood et al. (2017) as "shale" are not in fact not for shale gas that is released through high-volume hydraulic fracturing, but rather again for methane that has migrated from shales. In some cases, it is possible to determine from the original papers cited whether or not the samples are truly for shale gas, but in many cases thisis not possible. My response is to only use data for samples that unambiguously came from shale gases, and that clearly were not from

migrated gases." I have carefully looked at the data set of Sherwood et al., and I concur with this assessment by Howarth: the data set provides very little context on the data it includes. It would appear that samples are listed as being for "shale gas" if they come from a paper that uses the term "shale gas" in the key words, title, or abstract. One needs to go back to the original studies cited to determine if in fact these are for methane from actual produced shale gas: for the most part, they apparently are not or it is ambiguous whether they are or not.

Line 31: the comment asserts that the data used in Howarth (2019) are not representative of shale gas. This should be deleted, since the comment does not in fact establish this point.

Lines 34-35: the comment states "The Bakken is not a shale-gas play as clearly stated in the first sentence of Schoell et al. (2011), which the subject paper designates as shale-gas." Indeed, Howarth (2019) refers to the Bakken as a shale-gas play. In doing so, Howarth (2019) is entirely consistent with the Energy Information Agency of the US Department of Energy and the International Energy Agency, on which Howarth relied for data on shale gas production over time. See for example https://www.eia.gov/energyexplained/natural-gas/where-our-natural-gas-comes-from.php Howarth (2019) very clearly defined shale gas as the natural gas that is produced from shale formations through the use of high-volume hydraulic fracturing. In his comment, Lewan is apparently defining shale gas in some other way, although he does not clearly state his definition.

Lines 39 to 44: the comment states "Howarth (2019) gives a mean Bakken 13C value of -47.0‰ and cites Schoell et al. (2011) as one of the three data sets averaged to obtain the mean shale-gas value (i.e., -46.9‰. However, Schoell et al. (2011) report 13C values for only 8 gases that are associated with produced oil from the Bakken Shale at various levels of thermal maturity. All eight of these gases have 13C values heavier than -47.0‰ which makes the cited mean unattainable. A calculated mean of these eight values is -46.0 ± 0.74‰ (Table 1)." I have looked at Schoell et al. (2011),

and they present their data only in a figure; to my eye, Lewan is wrong when he states that all of the values are heavier than -47. He apparently got his estimate from values presented in the Sherwood et al. data set, and not from the original source. Beyond that, let's assume Lewan is right and Howarth (2019) should have used a value of -46 rather than -47: that changes the mean for all three shale plays in Howarth (2019) from -46.9 to -46.6; this has very little influence on the analysis or conclusions of Howarth (2019). The Lewan comment is nitpicking.

Lines 46 to 59: in the comment, Lewan writes "Botner et al. (2018) do an excellent job of reporting on the 13C values of methane dissolved in ground waters in Ohio where hydraulic-fracturing of the Utica Shale is occurring. They prescribe two 13C1 values for natural gas (conventional and shale gas in their Figure 3) for a visual comparison showing that the methane dissolved in the ground waters is independent of hydraulic-fracturing activity. Presumably, their conventional gas value of -41.3‰ was collected from an abandoned gas well through an intersecting water well, but the source of their shale gas value of -47.3‰ is not given." I looked up the Botner et al. (2018) reference, and Lewan is wrong. Botner et al. clearly state that they were reporting on samples taken from actual producing wells, for both the shale gas and conventional gas. They also clearly show that the shale gas well was more depleted in 13C, giving a 13C value of -47.3 for methane from a producing shale gas well and -41.3 for a producing conventional gas well.

In the comment, Lewan then goes on: "It is this single 13C value that Howarth (2019) uses to calculate his mean shale-gas value (i.e.,-46.9‰. This is unfortunate in light of the extensive 13C data reported by Burruss and Laughey (2010) on mostly unconventional gas sourced by the Utica Shale in the Appalachian region. These authors classified the collected gases with respect to whether they were associated or unassociated with oil production. Summary of these 55 analyses in Table 1 differentiated between the two with one gas produced from the Utica "proper" with a 13C value of -27.0‰ and 39 gases that have migrated out of the Utica into adjoining rock units with

a mean 13C value of -30.9‰Àlso given, is a mean 13C value of -38.7‰ for 23 gases associated with oil production at lower thermal maturities. These heavier values for the Utica shale gas do not support the prescribed single Utica value used by Howarth (2019; -47.3‰." I looked up the Burruss and Laughey (2010) paper, and Lewan has seriously mischaracterized these samples. Burrus and Laughey (2010) wrote "Most of the lower Palaeozoic gas samples we discuss in this paper were collected from unconventional fractured carbonate and tight sandstone reservoirs of Ordovician and Silurian age. However, three gas samples were collected from thermogenic shale gas producing zones in the Ordovician Utica and Point Pleasant shale source rocks." That is, only 3 of the 55 samples have anything at all to do with shale, and it is not clear whether or not these 3 samples represent shale gas as defined by Howarth or if they are samples for methane that had migrated from a shale formation.

Lines 61-65: the comment states "The Barnett data set from Townsend-Small et al. (2015) used in the subject paper is also inappropriate because the mean 13C value of -46.5 ‰ includes only atmospheric samples and not well-head samples of produced shale gas that have a mean value of -41.0 $\pm$ 2.6‰ (132 samples, Zumberge eta al., 2012). It is this isotopically heavier (12C depleted) mean value from well-heads collected over five counties that should be used as an example of Barnett shale gas and not atmospheric-gas samples from gas-well pads that can contain atmospheric microbial methane." Having read both this Lewan comment and Howarth (2019), I strongly disagree with the comment. Note that Howarth (2019) had explicitly presented the difference between the air sample from Townsend-Small et al. and the samples from Zumbergre et al., stating "Note that in the Barnett shale region, Texas, the $\delta$13C ratio for methane emitted to the atmosphere ($-46.5$; Townsend-Small et al., 2015) is more depleted than the average for wells reported in the Sherwood et al. (2017) data set: $-44.8$‰ for "group 2A and 2B" wells and $-38.5$‰ for "group 1" wells (Rodriguez and Philp, 2010) and a $-41.1$‰ average value (Zumberge et al., 2012)." Howarth (2019) also noted "In other cases, the data appear to come both from conventional vertical wells and shale-gas horizontal wells in the same region, making interpretation ambiguous (Rodriguez and Philp, 2010; Zumberge et al., 2012)." In short, Howarth (2019) chose to use the sample from Townsend-Small et al. because it was clearly coming from a producing shale gas well; the data from Zumberge et al. (and other studies noted by Howarth) may or may have been from shale gas at all, and were not from producing wells.

Further, as noted above, shale gas development may result in the release to the atmosphere of methane than includes very isotopically methane from biological sources, which can be vented from the well casings for the shale gas (https://agu.confex.com/agu/fm19/meetingapp.cgi/Paper/520383). This was not considered in Howarth (2019), which suggests that Howarth may have been too conservative in his assumptions: the methane released to the atmosphere from shale gas development may be even greater than the Howarth (2019) paper concluded.

Lines 66- 75: the comment states "the futility of the notion by Howarth (2019) that 13C values of shale gas are lighter than conventional gas is also shown in the Barnett gas data reported by Rodriguez and Philp (2010). They characterize their gases into two groups. Group 1 gases are methane-rich (>95% C1) and occur in the eastern more thermally mature part of the Fort Worth Basin (>1.2%Ro). Group 2 gases are wet (93 to 79% C1) and occur in the western less thermally mature part of the basin (<1.2%Ro). As expected and shown in Table 1, the mean 13C for the more thermally mature Group 1 gases (-38.5 $\pm$ 0.8‰ is heavier than that of the less thermally mature Group 2 gases (-44.5 $\pm$ 2.3‰. Like the Barnett gases reported by Zumberge et al. (2012), they are proper shale gases that have not experienced migration out of their tight host rock. Figure 1 shows both groups have heavier 13C values than that prescribed by Howarth (2019) and that thermal maturity and not migration are responsible for 75 their differences." I looked up the Rodriguez et al. reference, and the comment by Lewan is not accurately describing their study. Rodriguez and Philp give very little information on their samples, stating simply that they were provided by Devon Energy and that they included both horizontal and vertical wells; the horizontal wells would

presumably be for shale gas, but the vertical wells would not be shale gas. Rodriguez and Philp provide no data that would allow the separation of the C13 value from shale vs. conventional gas. And as noted above, the samples from Zumberge et al. also do not clearly indicate whether they come from producing shale gas wells.

Lines 76-88: the comment states "in addition to the Barnett being an unequivocal example of a shale-gas host, the Fayetteville of the Arkoma Basin and Marcellus of the Appalachian Basin are also unequivocal examples of major shale-gas hosts. Zumberge et al. (2012) report 13C values for shale gas produced from 98 wells in the Fayetteville over 5 counties in Arkansas with a mean of -38.2 ± 1.5‰." As discussed above and as noted in Howarth (2019), it is not at all clear which if any of the samples measured by Zumberge et al. come from producing shale gas wells, as opposed to conventional gas wells (where the methane had previously migrated from the shale) or from overly mature shales that had methane that is enriched with 13C but that might not represent commercially viable wells.

The comment goes on to state "similarly, the mean of 1,502 shale gases from mud-gas logging (MGL) in the Marcellus proper also have a significantly heavier isotopic signature with a mean 13C of -32.4 ± 3.8‰ (Table 1). It should be noted that gases from mud logging (MGL) have good one-to-one correlations with produces gases (Weissenburger and Borbas, 2004; Dawson and Murray, 2011)." I looked up these two references, one of which is a book chapter from 2004 before there was any significant shale gas development anywhere in the world (and therefore is really only talking about conventional gas); the other is an abstract from 2011, and provides very little detail. There is no way to tell if these samples represent producing shale gas wells.

Further, the comment states "Baldassare et al. (2014) present 682 13C values collected during mud-gas logging that represent shale gas from overlying rock units sourced by the Marcellus." This indicates Lewan's confusion in writing his comment: as he states here, most of the samples in Baldassare et al. are for methane that has migrated from the shale formations, and therefore do not represent shale gas production.

Lines 88-90: the comment states "the migration fractionation of 13C proposed by Howarth (2019), it is not supported by experimental data as reported by Zhang and Krooss (2001). They state that diffusion of methane through water-saturated sedimentary rocks is most likely to cause fractionation of 13C during migration, but diffusion is not a major mode of gas migration in hydrocarbon systems. Their experiments at subsurface conditions showed that the 13C of diffused methane is lighter than that of the methane source and not heavier as advocated by Howarth (2019)." I looked up the Zhang and Kroos paper, and it only addresses the influence of diffusion per se, and diffusion through saturated media at that. Howarth (2019) hypothesized a fractionation due to oxidation of methane as it migrates through formations such as sandstones over millions of years. The findings of Zhang and Kroos are not applicable to this hypothesis.

Lines 98-105: the comment states "Howarth (2019) speculates that conventional methane becomes isotopically heavier (13C enriched) during migration as a result of 12C being preferentially oxidized by "perhaps" bacteria using ferric iron or sulfate as the "oxidizing power". Unfortunately, the references he cites in this regard are not relevant. The papers by Whelan et al. (1989) and Rooze et al. (2016) are respectively concerned with anaerobic incubated shallow mud cores and near surface sediments with liable organic matter, and not subsurface rocks with kerogen." I guess "shallow" is subject to interpretation, but the Whelan et al. paper documented active sulfate reduction 167 m deep into cores. More importantly, Rooze et al. demonstrated that methane can be oxidized using Fe(III) as an electron acceptor. This shows the potential for methane to be oxidized during migration through sandstones, if the sandstones contains Fe (III), and therefore seems highly relevant to the hypothesis of Howarth (2019). That the Lewan comment refers to kerogen seems besides the point: the argument of Howarth is that methane is the organic matter being oxidized.

The comment goes on to state "The cited papers by Burruss and Laughrey (2010) and Hao and Zou (2013) discuss the possibility of ethane oxidation and do not consider

oxidation of methane." While these papers focus on oxidation of ethane (and larger alkanes), their findings of active oxidation-reduction chemistry involving Fe and sulfate are highly relevant to methane oxidation, and Hao and Zou (2013) specifically say "gases may have migrated out of the source rocks and probably undergone alteration by thermochemical sulfate reduction," referring to the study by Tilley et al. (2011).

The comment goes on further to state "Howarth (2019) does not note that Hao and Zou (2013) state that methane is the most stable petroleum compound and is not likely to be oxidized in the subsurface." Hao and Zou make this statement only in the context of methane that remains in a shale formation, which is a highly reducing environment. Methane is of course the most reduced carbon compound that exists, and so yes, is the most stable compound in a highly reducing system. However, methane is not stable in an environment that contains Fe(III) or sulfate, which can serve as electron acceptors to oxidize the methane. This is the context of the hypothesis of Howarth (2019); methane may be oxidized as it migrates away from the shale through sandstones to form a conventional gas reservoir.

Lines 106-115: the comment states "it is important to realize that 13C values are not a reliable parameter to differentiate shale gas from conventional gas on a global basis. As correctly stated and referenced by Howarth (2019), some shale gases have lighter 13C values (-50.7 to -53.3‰ but are not typical of major shale-gas plays. As shown in Figure 1, data from the cited references 110 have lighter 13C values (Martini et al., 1998; McIntosh et al., 2002; Osborn and McIntosh, 2010), which is attributed to the addition of microbial methane in shales that are at shallow depths or low thermal maturity (<1.2 %Ro) within sedimentary basins. Theoretically, 13C values for shale gas can span the full range of values observed for conventional gases as reported by Jenden et al. (1993) in Figure 1. However, major economic accumulations of shale gas occur in high thermal maturity host rocks that have heavier 13C values." The Howarth (2019) paper clearly states that there is a large range in the 13C content of methane from conventional natural gas, and likely from shale gas as well. The argument made

here by Lewan in his comment on thermal maturity would seem to apply equally to shale gas and conventional natural gas. The essential argument is whether or not the mean value for natural gas produced from shale wells is more depleted in 13C than the mean value from conventional gas wells. Howarth (2019) hypothesized that it is, because of methane oxidation during migration from the shale formations to conventional reservoirs.

Lines 116-119: the comment states "it should be noted that direct measurements of U.S. methane emissions between 2006 and 2015 by Lan et al. (2019) indicate that despite an ∼46% increase in gas production during this time period, total US methane emissions have remained essentially constant." I looked up the Lan et al. paper, and this characterization by Lewan is quite misleading. Lan et al. state that within the variance of the monitoring data they analyze, one cannot conclude there has been a major change in total methane emissions from all sources over the past decade. They go, however, and state that the monitoring data suggest an increase in methane emissions from oil & gas activities in the order of 3.4% per year (plus or minus 1.4%). This means that their best estimate is that methane emissions from oil & gas in the US increased by 35% over the 2006-2015 period, and by perhaps as much as 52% (based on 3.4% + 1.4% increase in emissions per year, compounded). This is not at all inconsistent with the conclusions of Howarth (2019).

I note that Howarth (2019) refers to Turner et al. (2016), a paper that used satellite data to infer that 30% to 60% of the total increase in methane emissions globally over the 2005-2015 time period came from the United States. In his comment, Lewan ignored this finding. He should at least acknowledge this analysis, and perhaps try to rectify it with the Lan et al. paper. (note that Lan et al. was only published in late April 2019, which may explain why it was not discussed in Howarth 2019).

Lines 121-131: the comment states "Table 1 gives the 13C means for compiled "proper" (-36.9 ± 6.3‰ and "proper plus migrated" (-36.5 ± 6.0‰ shale gases. Similar to the approach of Howarth (2019), these means are not weighted by number of samples

and are referred to as unweighted. These two means are essentially the same but are significantly heavier than the 13C mean of -46.9 ‰ shale-gas value prescribed by Howarth (2019). These heavier 13C values for shale gas are not unique to the U.S. (Table 1 and Figure 1) with major shale gas from the Chinese Longmaxi Shale also having heavy 13C values (-29.2 ± 1.2‰ mean of 76 samples; Feng et al., 2017)." It is not entirely clear what Lewan means by "proper" and "proper plus migrated"; nonetheless, it is interesting that methane that migrated from the shale (if this is what Lewan means) appears to be more depleted in 13C; this is consistent with the logic of Howarth (2019). Beyond this, it is questionable whether the data presented in Table 1 represent methane from producing shale gas wells, for these reasons articulated above. The Chinese data also seem questionable, and it is curious Lewan even metions these, since as of 2015 (the end of the time period analyzed by Howarth 2019), there had been absolutely no commercial shale gas development in China.

Lines 150-152: the comment states "interestingly, an increase in biogenic methane emission with its isotopically lighter 13C would better explain the decrease in atmospheric 13C since 2009 (Schaefer et al., 2016)." This statement ignores two more recent papers, both discussed in Howarth (2019), that very much undercut the conclusion of Schaefer et al. One of these, Schwietzke et al. (2016) used what they called an improved data set of 13C sources, and concluded that fossil fuel emissions are more important. The other, Worden et al. (2017), pointed out that Schaefer et al. had very much underestimated fossil fuel emissions and overestimated biological emissions by mischaracterizing changes in biomass burning.

Lines 146-151: It is rather amazing, after reading the extremely critical language throughout the Lewan comment, to see that his reanalysis using what he believes are better 13C values for shale gas results in estimated changes in global methane fluxes that are in fact not that different from the mean values presented in Howarth (2019): Lewan gives a value of 6.8 Tg per year for shale gas, compared to 9.4 Tg per year in Howarth. His reanalysis indicates that increases in total fossil fuel sources over

the past decade (13.7 Tg per year) are about the same as the increase in biological sources (14.7 Tg per year), while Howarth estimated increases of 17.6 Tg per year from total fossil fuels and 10.6 Tg per year from biological sources. In contrast, the papers by Schaefer et al. (2016) and by Schwietzke et al. (2016) – both of which Lewan treats rather uncritically in his comment – concluded that virtually all of the increase in methane emissions came from biological sources, and suggested that fossil fuel emission may have actually declined.

Lines 157-164: the comment has a rather long and dismissive sentence here, which I break down into these pieces: "Howarth (2019) does not use representative shale-gas isotopic data....." Lewan has failed to support this statement, as detailed above.

".....excludes a plethora of publicly available shale-gas data...." Again, I believe Howarth (2019) well explained why he did not use the data in question, as stated above.

"...... does not realize shale gas and conventional gas on a global basis cannot be readily distinguished based solely on 13C values....." I read Howarth (2019) at least in part as a cautionary message to those who overly rely on trends in 13C values to interpret trends in methane emissions, so this criticism by Lewan seems unfair.

"...... speculates contrary to field observations and laboratory experiments that migration causes conventional gases to have heavier 13C values than shale gas....." Lewan's evidence on this simply are not convincing, and the hypothesis of Howarth (2019) that fractionation can occur as methane is oxidized during migration through sandstone over millions of years sounds at least possible.

"..... does not consider the effects of thermal maturation on shale-gas 13C values....." This is not true. Howarth (2019) notes the importance of thermal maturation. But is there any reason to believe that this influences shale gases differently that conventional gas, that migrated from shale over time? If so, Lewan has not even tried to make that case.

[Figure]

".…... and neglects 13C data showing major shale-gas production is heavier and not lighter than conventionally produced gas. These numerous and significant shortcomings render his conclusions on global methane emissions from shale gas invalid." These assertions simply do not stand up to close scrutiny, as detailed above.

Table 1: as discussed above, the data in this table do not unambiguously come from samples of producing shale gas wells, and in many cases may instead come from gas that has migrated to conventional reservoirs. The table should be deleted.

---

## Referee Comment (RC2) · Anonymous Referee #2 · 3 Apr 2020

General This paper is a rejection of the proposal by R.D. Howarth (2019, cited by Lewan) that in the past decade, shale gas extraction in North America has been the main driver of increased global methane emissions from fossil fuels, and a major factor in the total growth of the atmospheric methane burden. Howarth's argument hinges on the isotopic data, asserting that emissions from shale gas extraction can drive the methane burden towards lighter, more negative $\delta13CCH4$ values. Lewan attacks this assertion by presenting an assortment of detailed data from many US gasfields, and then concludes that these numbers show that the $\delta13CCH4$ values of shale gas are typically heavier than those of conventional gas. This is the opposite of Howarth's conclusion. In particular, Lewan finds the mean shale gas $\delta13CCH4$ a little heavier than

[Figure]

-37‰ markedly heavier than values near -47‰ taken by Howarth. There is relevant new information available, published very recently in the well-argued paper by Milkov et al. (2020). In contrast to the scattered data used by both Howarth and Lewan, Milkov et al. (2020) construct a volume-weighted estimate of emissions, finding a volume-weighted average $\delta 13CCH4$ of $-39.6$‰ for US shale gas extracted since 2008. This value is not far from Lewan's estimate, but made from a much stronger database and more rigorous methodology than used by either Lewan or Howarth. From this, Milkov et al (2020) conclude that the "increase in global atmospheric CH4 is not dominated by emissions from shale gas and shale oil developments." Lewan cites Lan et al. (2019), who found little evidence for growth in North American methane emissions over the past decade. Note that Bruhwiler et al. (2018), also found that North American CH4 emissions in 2000-2012 have shown little change. These two papers, taken together with Milkov et al. (2020), collectively provide compelling evidence against Howarth's hypothesis. Thus, while Lewan's conclusion is not far distant from the findings of Milkov et al (2020), the detail and methodology of the Milkov et al (2020 approach is significantly superior. Thus, my recommendation is that while the broad conclusions are likely valid, this comment paper should be returned for significant revision and any resubmitted version should fully take into account Milkov et al. (2020). Specific Line 60. Lewan comments that the work by Townsend-Small (2015) is inappropriate because the finding of a $\delta 13CCH4$ value of -46.5 ‰ includes only atmospheric samples and ignores well-head gas. But it is exactly the emission that reaches the air that is crucial to the discussion! Townsend-Small's paper is important as it is a direct measurement.

REFERENCES Bruhwiler, L.M., Basu, S., Bergamaschi, P., Bousquet, P., Dlugokencky, E., Houweling, S., Ishizawa, M., Kim, H.S., Locatelli, R., Maksyutov, S. and Montzka, S., 2017. US CH4 emissions from oil and gas production: Have recent large increases been detected?. Journal of Geophysical Research: Atmospheres, 122(7), pp.4070-4083. Lan, X. et al. (2019) Long-term measurements show little evidence for large increases in total U.S. methane emissions over the past decade. Geophys. Res. Lett. 46, 4991–4999. Milkov, A. V., Schwietzke, S., Allen, G., Sherwood, O.

A., & Etiope, G. (2020). Using global isotopic data to constrain the role of shale gas production in recent increases in atmospheric methane. Scientific Reports, 10(1), 1-7.

Please also note the supplement to this comment:
https://www.biogeosciences-discuss.net/bg-2019-419/bg-2019-419-RC2-supplement.pdf

---

## Author Comment (AC1) · 20 Aug 2020

**Reply to Anonymous discussion author regarding Comments on "Ideas and perspectives: is shale gas a major driver of recent increase in global atmospheric methane?" by Robert Howarth (2019)**

By Michael D. Lewan, USGS Denver CO (mlewan@USGS.gov) Michael Whiticar, University of Victoria, CA (whiticar@uvic.ca) Alexei Milkov, Colorado School of Mines CO (amilkov@mines.ed)

Considering the high radiative forcing of atmospheric methane, it is important to reliably identify and quantifying the sources and sinks to understand the methane budget. The challenge is further emphasized by the renewed increase in net methane accumulation in the atmosphere since 2007, and identifying the cause(s) for the changes. The assignment of representative stable isotope ratios, e.g.,  $\delta^{13}$ CH4, to the various methane sources and isotope fractionation by sinks is an integral part of this methane budget process. If we are to effect real reductions in methane emissions, then accurately quantifying the flux strengths and signatures is a requisite need.

The Howarth (2019) paper, which has precipitated this discussion, assigned  $\delta^{13}$ CH4 values of shale gas ('SG'), conventional natural gas (termed 'CG'), gas associated with oil production, gas associated with coal production, and biogenic (microbial, 'B') and biomass burning (pyrogenic) emitting to the atmosphere (Howarth's Table 1). We argue that the  $\delta^{13}$ CH4 values for SG chosen by Howarth (2019) are demonstrably not representative. As these values cascade through the calculated mass balance in Howarth (2019), this leads to erroneous conclusions. The selection of more representative  $\delta^{13}$ CH4 values for fossil fuels and the attendant supporting information is provided and discussed in Lewan (2020). This paper responds directly to the comments in the discussion by the Anonymous Referee #1 (AR1, 2020).

The issue revolves largely around the selection of the  $\delta^{13}$ CH4 values for shale gas (SG). First, Lewan (2020) commented that there is substantial overlap in the range of  $\delta^{13}$ CH4 values for shale methane and conventional methane. This ineffectiveness of  $\delta^{13}$ CH4 values to unambiguously differentiate sources has been stated by numerous authors, e.g., Nisbet et al. (2016), Turner et al. (2019). The  $\delta^{13}$ CH4 overlap and lack of discrimination between SG and CG has been clearly shown by the extensive global natural gas database from Milkov et al. (2020). The latter paper further shows that the range in  $\delta^{13}$ CH4 values for SG and CG is large (46.7 ‰ and 88.2 ‰, respectively). However, their mean  $\delta^{13}$ CH4 values for SG and CG are respectively -41.1 ‰ and -42.8 ‰, which are notably different from Howarth (2019) values of -46.9 ‰ and -44 ‰, respectively. Although AR1 (2020) refers to these concerns by Lewan 2019 as 'nit picking', having representative  $\delta^{13}$ CH4 values is important as we will demonstrate.

Lewan (2019) employed the same rational and mathematical constructs as in Howarth (2019), but with  $\delta^{13}$ CH4 values deemed more representative and appropriate for shale gas. As a result, the revised calculation shows that shale gas is not the main driver of recent increases in global methane atmosphere. Whereas opposed to percentages, actual magnitudes are very important with shale gas methane being reduced by 2.6 Tg/year and all other fossil-fuel methane reduced by 1.5 Tg/year for a total fossil fuel methane reduction of 4.1 Tg/year with a calculated biogenic methane increase of 4.1 Tg/year, which is 14.4% of the 28.4 Tg /year global increase determined by Worden et al. (

2017). As noted by AR1 (2020), these more representative isotopic values change the conclusion prescribed by Howarth (2019).

The definition of shale gas used by Howarth (2019), as produced methane and associated gases released from shale rock by high volume hydraulic fracturing, is not universally accepted. In some instances, hydraulic fracturing is not a requirement and that shale gas is termed as natural gas produced from shales (e.g., Curtis, 2002; Jarvie, 2012). Regardless, of which definition is preferred, the database compiled by Lewan (2019, Table 1) complies with the definition of Howarth (2019), albeit some of the AR1 (2020)'s and those of Howarth (2019) examples are not consistent with the definition of Howarth (2019).

The AR1 (2020) uses the study by Muehlenbachs and Arismendi (2019) in the Western Canada Sedimentary Basin (WCSB) as evidence for the isotopically light shale gas methane emissions. In the abstract of this study, the gases reported are from surface casing vent flow and ground migration gases that have 13C-depleted (isotopically 'light')  $\delta^{13}$ CH4 values (-58.4 and -57.3%). respectively) suggesting a major microbial (biogenic) methane component. According to the authors, these emissions originate mostly from formations shallower than the shale gas horizons, which is indicative of migrated gases not related to the shale gas in underlying targeted shale intervals. It is common for groundwaters, especially under anaerobic conditions, to contain microbial methane generated via methanogenesis. In addition to  $\delta^{13}$ CH4 values, there is no supporting evidence presented by AR1 (2020) demonstrating that these gases were originally thermogenic shale gases. Therefore, it is tenuous solely on the Muehlenbachs and Arismendi (2019) study to state that shale gas is more 13C-depleted than conventional gas. In fact, stray gas and ground water gases are typically from microbial sources with 13C-depleted  $\delta^{13}$ CH4 values (e.g., Jackson et al., 2013; and Botnar et al., 2018; and Kulongoski, et al., 2018). These results do not provide unequivocal evidence that shale gas development is responsible for the increase in atmospheric methane emissions over the past decade.

**Specific: detailed responses:**

Lines 10-12: The recent paper by Milkov et al. (2020) that compiles over 1,656 global SG samples and 12,416 CG samples is considered to be more robust than the 43 selected SG samples used by Howarth (2019). This extensive data base by Milkov is now included in the revised Comment paper.

Lines 21-24:The Comment by Lewan (2020) should have included the work and conclusions by Worden et al. (2017). The work by Worden et al. (2017) is cited by Lewan (2019) but within the context of balancing increases of 28.4 Tg/yr of methane emissions as used in Equation 1 presented by Howarth (2019). Similarly, the failure to cite Turner et al. (2016) was simply an oversight.

Lines 26-31: The comment by Lewan (2019) was intended to gain clarity on gaining an explanation as to why the Howarth (2019) did not utilize the Sherwood et al. (2017) database and eliminate the entries he did not consider to be shale gas. Howarth (2019) or AR1 (2020) to evaluate the  $\delta^{13}$ CH4 shale gas from horizontal wells versus so called conventional gas from vertical wells would have provided a great comparison for Howarth (2019) to use in his subject paper to document their claims. A similar concern arises with Howarth's (2019) argument that  $\delta^{13}$ CH4 values of shale gas are different from migrated conventional gas.

Lewan (2019) consciously did not include the data by Tilley and Muehlenbachs (2013) to ensure omission of migrated CG from his data set and only used proper shale gas entries to determine a mean value. Certainly, the use of datasets requires due diligence and scrutiny to ascertain their reliability. If the data selection is flawed, then it is important to determine how this impacts the conclusions. Yes, an investigator does need to go back through a data set to make determinations as to whether the  $\delta^{13}$ CH4 values are from shale gas or conventional gas. And yes, this may require going back to the original papers to establish their type. This is the best practice before condemning other large data bases as being unusable. So, if some of the data in a data set were as stated by Howarth (2019) unreliable, then clear examples should be shown how the inclusion of these questionable values impact the outcome.

Lines 34-35: Lewan (2019) used the strict definition by Howarth (2019) of shale gas and used the term "proper" to emphasize the gases that are produced from shale. The Bakken gasses associated with oil production in the Bakken Shale Formation are actually produced from clastic and carbonate reservoirs in the middle of the Bakken Shale Formation and not from the over or underlying shale lithologies. In this case, the gases are migrated and senso stricto not shale gases because of their migration according to the definition prescribed by Howarth (2019). Therefore, the oil associated gases in the Bakken should not be included in the mean of shale gas  $\delta^{13}CH_4$ values. The AR1 (2020) asserts that Lewan (2019) was mistaken in stating that all of the Bakken gas isotope values plotted by Schoell and LeFever (2011) are heavier than -47‰. As shown in Figure 1 from Schoell and Lever (2011), the Bakken gases are heavier than -47‰. The values inserted in this plot are taken from Schoell and Lever (2011) and not from Sherwood et al. (2017) as suggested by AR1 (2020). It should be noted that Figure 1 is from a different article than cited by Howarth (2019), i.e., Schoell et al. (2011). But the plots in this latter paper have the same gases with  $\delta^{13}$ CH4 values more 13C-enriched ('heavier') than -47.0 ‰. Thus, the proposed mean for the Bakken gases by Howarth (2019) is not possible. The  $\delta^{13}$ CH4 values in Schoell et al. (2011) are also heavier than -47 ‰ with values between -44.97 ‰ and -46.92 ‰ and a mean of -46.0 ‰, similar to Figure 1. Although the difference in the choice of a mean of -47‰ by Howarth (2019) is not consequential in the overall atmospheric mass balance, it clarifies the difference in Bakken values by Howarth and Lewan in Table 1.

Figure 1. Plot presented by Schoell and LeFever (2011) showing composition and  $\delta^{13}$ CH4 values for Associated gases produced with oil from the Bakken Shale.

Lines 46 – 59: The AR1 (2020) correctly notes that Botner et al. (2018) obtained their conventional and shale gases from production wells, although the actual well data or statistics were not presented in the paper. A personal communication with Botner (April 17, 2020) confirms these two values represent single gases collected from a producing SG and CG gas wells in Carroll County and are not a statistical mean of several produced gases from the Utica. As stated by Lewan (2019) and Botner et al. (2018), the gas representing the CG is from a residential groundwater well that drilled into a conventional gas well. The SG sample was taken downwind from a shale gas well and corrected for atmospheric air. The thermal maturity of the shales in these two wells is not given, but thermal maturity maps indicate that Carroll County (Riley, 2016) straddles a range of thermal maturities from oil generation (0.6-0.8 % VRo) through wet gas (0.8-1.0% VRo). This can have a notable effect on the  $\delta^{13}$ CH4 values depending on the specific location of the wells and explain the differences in values without invoking oxidation during migration.

AR1 (2020) correctly states that the data set of Burruss and Laughery (2010) does not distinguish between shale and conventional gases. However, Burruss and Laughery (2010) do identify a gas produced from the Utica, which is considered a proper shale gas in Table 1 of Lewan (2019). The other gases are collectively grouped in the migrated gas category, but may include some shale gases. The posted Proper Utica shale gas  $\delta^{13}$ CH4 value of -27.0‰ is much heavier than the -47.3‰ used by Howarth (2019). Milkov et al. (2020) also posted a heavier mean  $\delta^{13}$ CH4 of -31.8‰ for the Utica based on 4 samples. It is worth noting that the total mean reported in Table 1 of Lewan (2019) for proper SG does not include any  $\delta^{13}$ CH4 values from Burruss and Laughery (2010) with the exception of the one sample produced from the Utica. Because the Burruss and Laughery (2010) data cannot be differentiated as SG or CG, 39 of their samples are classified as migrated shale gas in Table 1.

**Lines 61-65:**

It remains difficult to understand the rational of AR1 (2020) to represent the  $\delta^{13}$ CH4 for the Barnett Shale with atmospheric samples, which are known to be admixed with microbial gas component, instead of using the values of Zumberge et al. (2012) obtained from gases produced directly from the Barnett Shale. The relationship between the atmospheric gas samples from Townsend et al. (2015) to the produced samples reported by Zumberge et al. (2012) is not established. AR1 (2020) states that the Rodriguez and Philp (2010) data are from horizontal and vertical wells with no distinction between the two, which is correct. However, it is critical to state that these vertical and horizontal wells are all producing gas from the Barnett Shale, which makes them shale gas regardless of the drilling direction. It is useful to note that it is convention to denote horizontal wells with an upper-case **H** in the suffix of a well number. Consequently, 89% of the Zumberge et al. (2012) gases or shale gases are from horizontal wells. The Zumberge et al. (2012) data set for the Barnett Shale, using only gases from horizontal wells gives essentially the same statistical values as those in Table 1 with a mean  $\delta^{13}$ CH4 of -41.0 ± 2.4 ‰. Interestingly, the remaining 18 Barnett Shale gases from vertical wells have a mean  $\delta^{13}$ CH4 of -41.2 ± 3.1 ‰, which is indistinguishable from horizontally drilled gases. Therefore, the values given in Table 1 of Lewan (2019) to determine a mean shale gas  $\delta^{13}$ CH4 remain reasonable and representative of shale gas from the Barnett Shale, and are isotopically heavier than -46.5‰ used by Howarth (2019).

Shale gas can be produced from both vertical and horizontal wells, in contrast to the positions of Howarth (2019) and AR1 (2020). For example, Devonian shale gas has been produced in New York State from vertical unstimulated wells since 1821 (Curtis, 2002). Fayettville Shale data set (Zumberge et al., 2012) has a mean  $\delta^{13}$ CH4 of -37.4 ± 5.4 ‰, which is the best mean  $\delta^{13}$ CH4 for Fayetteville proper (horizontal wells) gases in Table 1 of Lewan (2020). This is statistically not significantly different than the mean  $\delta^{13}$ CH4 of -38.9 ±1.2 ‰ for the vertical gas wells, or for the mean  $\delta^{13}$ CH4 of -38.2 ±1.5 ‰ given in Lewan (2020) Table 1.

AR1 (2020) challenges, but without any supporting information, the use of the mud gas logging (MGL) isotope values for the Marcellus Shale (e.g., Baldassare et al., 2014). MGL is a wellestablished, industry technology, (e.g., McKinney et al 2007, Berman et al., 2002, Ellis et al., 1999, 2003, Stankiewicz, et al., 2007) that strongly supports the inclusion of MGL  $\delta^{13}$ CH4 data, including the Marcellus Shale. The AR1 (2020) argument using Dawson and Murray (2011) does not necessarily pertain to shale gas methane. Although it is an extended abstract, it includes figures with plotted data. Figure 2, shows an excellent one-to-one correlation between  $\delta^{13}$ CH4 values of bottom-hole gas, i.e., produced gas, and MGL gas. Whether these plotted gases are shale gas or conventional gas is not pertinent; it does demonstrate that MGL gas signatures accurately reflect that of produced gas.

---

## Author Comment (AC2) · 20 Aug 2020

Reply/reconciliation to Anonymous reviewer 2 (AR2)

Michael D. Lewan

This review is interesting in that it wanders around and requests major changes, but the specific major changes are not explicitly defined. Here is how I have broken down the AR2 review.

1. One obvious major change is the inclusion of Milkov's work, which has been done and he is now a coauthor of the revised Comment manuscript. It should be noted that the data in Milkov's paper (2020) was not publicly available at the time the original comment paper was written. As AR2 states, the Milkov data does not really change the conclusion of Lewan (2019) that the isotopic values used by Howarth (2019) are not representative. This is conveyed numerous times in the revised manuscript.

2. The inclusion of the Bruhwiler (2017) paper has not been included with the Lan reference as suggested. We simply used the most recent published NOAA report by Lan (2019).

3. There is no doubt that the Milkov data with weighted averages is a better data set than that proposed by Lewan (2019). Lewan (2019) better addresses the specific short comings of Howarth's data base and shows the resulting discrepancy that results in a more representative data base. This is also shown in the revised manuscript.

4. Regrettably, Towsend-Small atmospheric $\delta^{13}CH_4$ data has not demonstrated that the atmospheric methane originated from the gas production facilities. Until a connection is made, the difference of -5.5‰ between the produced and atmospheric methane (i.e., -46.5 and -41.0‰, respectively) needs to be established. The Keeling plots to correct the measured $\delta^{13}CH_4$ values show considerable scatter and as a result the corrections are questionable. It is interesting that the raw data presented in the supplemental data by Twonsend-Small show no significant difference among the up-wind (background) $\delta^{13}CH_4$ values (- 47.9± 0.2‰) and the downwind values (-47.8± 2.0‰) and the production pad values (-47.9± 2.0‰). A more unequivocal data set would take measurements from an established production leak with samples at various distances vertical and lateral from the leak. This type of data would establish the relationship between actual methane production leaks and what processes are operative as it becomes depleted in $^{13}C$ as it enters the atmosphere. Therefore, it does not appear to be necessary to side track the manuscript to content with Townsend-Smalls work. This further explanation of the problems with Town-send-Small's work is included in the revised manuscript and subsequent lines 60 will not be removed and can be defended.